# MVDream:
# Multi-view Diffusion for 3D Generation

**Yichun Shi**[1], **Peng Wang**[1], **Jianglong Ye**[2][*] **Long Mai**[1], **Kejie Li**[1], **Xiao Yang**[1]

[1] ByteDance, USA, [2] University of California, San Diego

`{yichun.shi,peng.wang,kejie.li,xiao.yang}@bytedance.com`
`{jianglong.yeh,mai.t.long88}@gmail.com`

## ABSTRACT

We introduce *MVDream*, a diffusion model that is able to generate consistent multi-view images from a given text prompt. Learning from both 2D and 3D data, a multi-view diffusion model can achieve the generalizability of 2D diffusion models and the consistency of 3D renderings. We demonstrate that *such a multi-view diffusion model is implicitly a generalizable 3D prior agnostic to 3D representations*. It can be applied to 3D generation via Score Distillation Sampling, significantly enhancing the consistency and stability of existing 2D-lifting methods. It can also learn new concepts from a few 2D examples, akin to DreamBooth, but for 3D generation. Our project page is `https://MV-Dream.github.io`

## 1 INTRODUCTION

3D content creation is an important step in the pipeline of modern game and media industry, yet it is a labor-intensive task that requires well-trained designers to work for hours or days to create a single 3D asset. A system that can generate 3D content in an easy way for non-professional users is thus of great value. Existing 3D object generation methods can be categorized into three types: (1) template-based generation pipeline, (2) 3D generative models, and (3) 2D-lifting methods. Due to limited accessible 3D models and large data complexity, both template-based generators and 3D generative models struggle to generalize effectively to arbitrary object generation. Their generated content is often confined to common real-world objects with relatively simple topology and texture. Yet in industry, popular 3D assets usually come as a mixture of complicated, artistic, and sometimes non-realistic structures and styles (Ske).

Recently, 2D-lifting methods have shown that pre-trained 2D generation models can be potentially applied to 3D generation. The typical representations are Dreamfusion (Poole et al., 2023) and Magic3D (Lin et al., 2023a) systems, which utilize 2D diffusion models as supervision for the optimization of a 3D representation via score distillation sampling (SDS). Trained on large-scale 2D image datasets, these 2D models are able to generate unseen and counterfactual scenes whose details can be specified through a text input, making them great tools for creating artistic assets.

Nevertheless, in 2D-lifting techniques, challenges arise due to the lack of comprehensive multi-view knowledge or 3D-awareness during score distillation. These challenges encompass: (1) The multi-face Janus issue: The system frequently regenerates content described by the text prompt. (2) Content drift across different views. Examples can be seen in Fig. (1). The multi-face issue can stem from various factors. For instance, certain objects, like blades, may be nearly invisible from some angles. Meanwhile, vital parts of a character or animal might be hidden or self-occluded from specific viewpoints. While humans assess these objects from multiple angles, a 2D diffusion model cannot, leading it to produce redundant and inconsistent content.

In spite of all the weaknesses of 2D-lifting methods, we believe that large-scale 2D data is crucial to generalizable 3D generation. Therefore, we propose multi-view diffusion models, which can be used as a multi-view 3D prior agnostic to 3D representations. The proposed model simultaneously generates a set of multi-view images that are consistent with each other. It can leverage pre-trained 2D diffusion models for transfer learning to inherit the generalizability. Then, by jointly training the model on multi-view images (from 3D assets) and 2D image-text pairs, we find that it can achieve both good consistency and generalizability. When applied to 3D generation through score distillation,

---

[*]Work done during internship at ByteDance

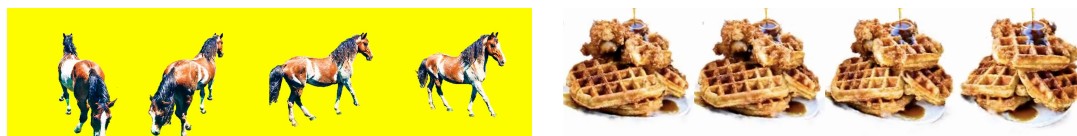

Multi-face Janus Problem          Content Drift Problem

Figure 1: Typical multi-view consistency problems of 2D-lifting methods for 3D generation. Left: "A photo of a horse walking" where the horse has two faces. Right: "a DSLR photo of a plate of fried chicken and waffles with maple syrup on them" where the chicken gradually becomes a waffle.

our multi-view supervision proves significantly more stable than that of single-view 2D diffusion models. And we can still create unseen, counterfactual 3D contents as from pure 2D diffusion models. Inspired by DreamBooth (Ruiz et al., 2023), we can also employ our multi-view diffusion model to assimilate identity information from a collection of 2D images and it demonstrates robust multi-view consistency after fine-tuning. Overall, our model, namely *MVDream*, successfully generates 3D Nerf models without the multi-view consistency issue. It either surpasses or matches the diversity seen in other state-of-the-art methods.

## 2   RELATED WORK AND BACKGROUND

### 2.1   3D GENERATIVE MODELS

The significance of 3D generation has driven the application of nearly all deep generative models to this task. Handerson *et al.* explored Variational Auto Encoders (Kingma & Welling, 2014) for textured 3D generation (Henderson & Ferrari, 2020; Henderson et al., 2020). However, their studies mainly addressed simpler models leveraging multi-view data. With Generative Adversarial Networks (GANs) yielding improved results in image synthesis, numerous studies have investigated 3D-aware GANs (Nguyen-Phuoc et al., 2019; 2020; Niemeyer & Geiger, 2021; Deng et al., 2022; Chan et al., 2022; Gao et al., 2022). These methods, attributed to the absence of reconstruction loss involving ground-truth 3D or multi-view data, can train solely on monocular images. Yet, akin to 2D GANs, they face challenges with generalizability and training stability for general objects and scenes. Consequently, diffusion models, which have shown marked advancements in general image synthesis, have become recent focal points in 3D generation studies. Various 3D diffusion models have been introduced for tri-planes (Shue et al., 2023; Wang et al., 2023b) or feature grids (Karnewar et al., 2023). Nevertheless, these models primarily cater to specific objects like faces and ShapeNet objects. Their generalizability to the scope of their 2D counterparts remains unverified, possibly due to 3D representation constraints or architectural design. It is pertinent to note ongoing research endeavors to reconstruct object shapes directly from monocular image inputs (Wu et al., 2023; Nichol et al., 2022; Jun & Nichol, 2023), aligning with the increasing stability of image generation techniques.

### 2.2   DIFFUSION MODELS FOR OBJECT NOVEL VIEW SYNTHESIS

Recent research has also pursued the direct synthesis of novel 3D views without undergoing reconstruction. For instance, Watson et al. (2023) first applied diffusion models to view synthesis using the ShapeNet dataset (Sitzmann et al., 2019). Zhou & Tulsiani (2023) extended the idea to the latent space of stable diffusion models (Rombach et al., 2022) with an epipolar feature transformer. Chan et al. (2023) enhanced the view consistency by re-projecting the latent features during diffusion denoising. Szymanowicz et al. (2023) proposed a multi-view reconstructer using unprojected feature grids. A limitation shared across these approaches is the boundedness to their respective training data, with no established evidence of generalizing to diverse image inputs. Liu et al. (2023) propose to fine-tune a pre-trained image variation diffusion model (sdv) on an extensive 3D render dataset for novel view synthesis. Notwithstanding, the synthesized images from such studies still grapple with geometric consistency, leading to a discernible blurriness in the output 3D models. Recently, Tang et al. (2023b) proposed a multi-view diffusion model for panorama with homography-guided attention, which is different from ours where explicit 3D correspondence is not available.

### 2.3   LIFTING 2D DIFFUSION FOR 3D GENERATION

Given the bounded generalizability of 3D generative models, another thread of studies have attempted to apply 2D diffusion priors to 3D generation by coupling it with a 3D representation, such as a

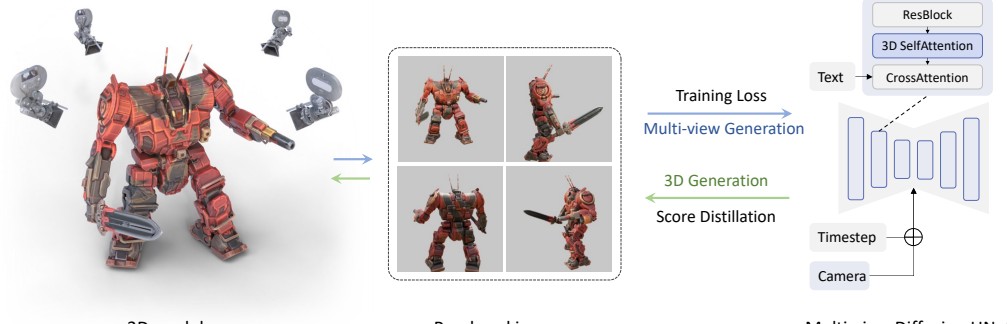

Figure 2: Illustration of the multi-view diffusion model. We keep the structure of text-to-image UNets by making two slight changes: (1) changing the self-attention from 2D to 3D for cross-view connection (2) adding camera embeddings for each view. Multi-view renderings are used to train the diffusion model. During testing, the pipeline is used in a reverse way: the multi-view diffusion model serves as 3D prior to optimize the 3D representation via Score Distillation Sampling (SDS).

NeRF (Mildenhall et al., 2021). The key technique of such methods is the score distillation sampling (**SDS**) proposed by Poole et al. (2023), where the diffusion priors are used as score functions to supervise the optimization of a 3D representation. Concurrent with Dreamfusion, SJC (Wang et al., 2023a) proposed a similar technique using the publicly available stable-diffusion model (Rombach et al., 2022). Following works have been further improving the 3D representations (Lin et al., 2023a; Tsalicoglou et al., 2023; Tang et al., 2023a; Chen et al., 2023), sampling schedules (Huang et al., 2023), and loss design (Wang et al., 2023c). Although these methods can generate photo-realistic and arbitrary types of objects without training on any 3D data, they are known to suffer from the multi-view consistency problem, as discussed in Section 1. In addition, as discussed in (Poole et al., 2023), each generated 3D model is individually optimized by tuning prompts and hyper-parameters to avoid generation failures. Nevertheless, in MVDream, we significantly improve the generation robustness, and are able to produce satisfactory results with a single set of parameters without individual tuning.

## 3 METHODOLOGY

### 3.1 MULTI-VIEW DIFFUSION MODEL

To mitigate the multi-view consistency issue in 2D-lifting methods, a typical solution is to improve its viewpoint-awareness. For example, Poole et al. (2023) add viewpoint descriptions to text conditions. A more sophisticated method would be incorporating exact camera parameters like in novel view synthesis methods (Liu et al., 2023). However, we hypothesize that even a perfect camera-conditioned model is not sufficient to solve the problem and the content in different views still could mismatch. For example, an eagle might look to its front from front view while looking to its right from back view, where only its body is complying with the camera condition.

Our next inspiration draws from video diffusion models. Since humans do not have a real 3D sensor, the typical way to perceive a 3D object is to observe it from all possible perspectives, which is similar to watching a turnaround video. Recent works on video generation demonstrate the possibility of adapting image diffusion models to generate temporally consistent content (Ho et al., 2022; Singer et al., 2022; Blattmann et al., 2023; Zhou et al., 2022). However, adapting such video models to our problem is non-trivial as geometric consistency could be more delicate than temporal consistency. Our initial experiment shows that content drift could still happen between frames for video diffusion models when there is a large viewpoint change. Moreover, video diffusion models are usually trained on dynamic scenes, which might suffer from a domain gap when applied to generating static scenes.

With such observations, we found it important to directly train a multi-view diffusion model, where we could utilize 3D rendered datasets to generate static scenes with precise camera parameters. Fig. (2) shows an illustration of our text-to-multi-view diffusion model. We leverage the 3D datasets to render consistent multi-view images to supervise the diffusion model training. Formally, given a set of noisy image $\mathbf{x}_t \in \mathbb{R}^{F \times H \times W \times C}$, a text prompt as condition $y$, and a set of extrinsic camera parameters $\mathbf{c} \in \mathbb{R}^{F \times 16}$, multi-view diffusion model is trained to generate a set of images $\mathbf{x}_0 \in \mathbb{R}^{F \times H \times W \times C}$

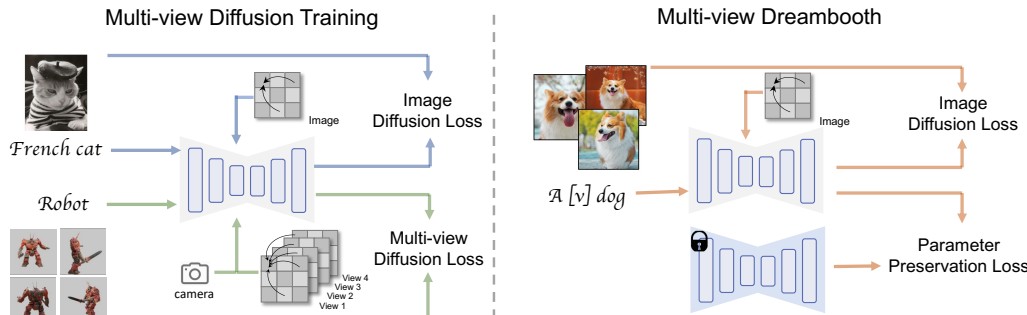

Figure 3: The training pipeline of MVDream. **Left**: Training of multi-view diffusion with two modes: image mode with 2D attention (upper) and multi-view mode with 3D attention and camera embeddings (lower). **Right**: Training of DreamBooth, where the pre-trained multi-view diffusion model is fine-tuned with the image mode of 2D attention and a preservation loss.

of the same scene from $F$ different view angles. After the training, the model can be used as a multi-view prior for 3D generation with techniques such as Score Distillation Sampling (SDS).

To inherit the generalizability of the 2D diffusion models, we would like to keep their architecture as much as possible for fine-tuning. However, such models can only generate one image at a time and do not take camera conditions as inputs. So the main questions here are: (1) how to generate a set of consistent images from the same text prompt (Sec. 3.1.1), (2) how to add the camera pose control (Sec. 3.1.2), and (3) how to maintain the quality and generalizability (Sec. 3.1.3).

### 3.1.1 MULTI-VIEW CONSISTENT GENERATION WITH INFLATED 3D SELF-ATTENTION

Similar to video diffusion models (Ho et al., 2022; Singer et al., 2022; Blattmann et al., 2023; Zhou et al., 2022), we would like to adapt the attention layers to model the cross-view dependency while keeping the remaining network as a 2D model that only operates within a single image. However, we found that a simple temporal attention fails to learn multi-view consistency and content drift still happens even if we fine-tune the model on a 3D rendered dataset. Instead, we choose to use a 3D attention. Specifically, we can inflate the original 2D self-attention layer into 3D by connecting all different views in self-attention (See Fig. 3), which we found able to generate rather consistent images even when the view gap is very large. Specifically, given a tensor of shape $B \times F \times H \times W \times C$, the we format it as $B \times FHW \times C$ for self-attention, where the second dimension is the sequence dimension representing the tokens. In such a way, we could also inherit all the module weights from original 2D self-attention. Note that we also experimented with incorporating a new 3D self-attention layer rather than modifying the existing 2D one. However, we found that such a design compromised the generation quality of multi-view images.

### 3.1.2 CAMERA EMBEDDINGS

Like video diffusion models, position encoding is necessary for our model to distinguish between different views. To this end, we compared relative position encoding (Singer et al., 2022), rotary embeddings (Su et al., 2021), and absolute camera parameters. We found that embedding camera parameters with a 2-layer MLP leads to the most satisfying image quality with distinguishable view differences. Specifically, we consider two methods to inject camera parameters: (1) adding camera embeddings to time embeddings as residuals, and (2) appending camera embeddings to text embeddings for cross attention. Our experiment shows that both methods work but the former turns out to be more robust possibly because the camera embeddings would be less entangled with the text.

### 3.1.3 TRAINING LOSS FUNCTION

We found that the details in data curation and training strategy are important to the image generation quality as well. For the space limit, we refer the readers to the *Appendix* for the details on data processing. For the training, we fine-tune our model from the Stable Diffusion v2.1 base model (512×512 resolution), where we keep their settings for the optimizers and $\epsilon$-prediction but reduce image size to 256×256. We found that a joint training with a larger scale text-to-image dataset is

helpful for the generalizability of the fine-tuned model, as illustrated in the left Fig. (3). Formally, given text-image dataset $\mathcal{X}$ and a multi-view dataset $\mathcal{X}_{mv}$, for training samples $\{\mathbf{x}, y, \mathbf{c}\} \in \mathcal{X} \cup \mathcal{X}_{mv}$ ($\mathbf{c}$ is empty for $\mathcal{X}$), the multi-view diffusion loss is defined as

$$\mathcal{L}_{MV}(\theta, \mathcal{X}, \mathcal{X}_{mv}) = \mathbb{E}_{\mathbf{x}, y, \mathbf{c}, t, \epsilon}\Big[\|\epsilon - \epsilon_\theta(\mathbf{x}_t; y, \mathbf{c}, t)\|_2^2\Big] \tag{1}$$

where $\mathbf{x}_t$ is the noisy image generated from random noise $\epsilon$ and images $\mathbf{x}$, $y$ is the condition, $\mathbf{c}$ is the camera condition and the $\epsilon_\theta$ is the multi-view diffusion model. In practice, with a $30\%$ chance we train the multi-view model as a simple 2D text-to-image model on a subset of LAION dataset (Schuhmann et al., 2022) by turning off the 3D attention and camera embeddings.

## 3.2 TEXT-TO-3D GENERATION

We recognize two ways to utilize a multi-view diffusion model for 3D generation:

- Using the generated multi-view images as inputs for a few-shot 3D reconstruction method.
- Using the multi-view diffusion model as a prior for Score Distillation Sampling (SDS).

Although more straightforward, 3D reconstruction requires a robust few-shot 3D reconstruction method, which is not available at the moment during this project. Therefore, we focus our experiments on the latter, where we modify existing SDS pipelines by replacing Stable Diffusion model with our multi-view diffusion model. This leads to two modifications: (1) changing the camera sampling strategy and (2) feeding the camera parameters as inputs. Instead of using direction-annotated prompts as in Dreamfusion (Poole et al., 2023), we use original prompts for extracting the text embeddings.

In spite that such a multi-view SDS can generate consistent 3D models, the content richness and texture quality still fall short of those images directly sampled by the denoising diffusion process. Thus, we propose several techniques to alleviate the issue. First, we linearly anneal the maximum and minimum time step for SDS during optimization. Second, to prevent the model from generating styles of low quality 3D models in the dataset, we add a few fixed negative prompts during SDS. Finally, to alleviate the color saturation from large classifier free guidance (CFG), we would like to apply clamping techniques such as dynamic thresholding (Saharia et al., 2022) or CFG rescale from (Lin et al., 2023b). Since these tricks only apply to $\hat{\mathbf{x}}_0$, we propose to use an $\mathbf{x}_0$-reconstruction loss instead of original SDS formulation:

$$\mathcal{L}_{SDS}(\phi, \mathbf{x} = g(\phi)) = \mathbb{E}_{t, \mathbf{c}, \epsilon}\Big[\|\mathbf{x} - \hat{\mathbf{x}}_0\|_2^2\Big]. \tag{2}$$

It can be shown that Eq. (2) is equivalent to the original SDS with $w(t)$, a hyper-parameter in SDS, equal to signal-to-noise ratio (See Appendix). Here $\mathbf{x} = g(\phi)$ refers to rendered images from 3D representation $\phi$ and $\hat{\mathbf{x}}_0$ is the estimated $\mathbf{x}_0$ from $\epsilon_\theta(\mathbf{x}_t; y, \mathbf{c}, t)$, whose gradient is detached. Empirically, we found that $\mathbf{x}_0$-reconstruction loss performs similarly to the original SDS but it mitigates the color saturation after we apply the CFG rescale trick (Lin et al., 2023b) on $\hat{\mathbf{x}}_0$.

We also use the point lighting (Poole et al., 2023) and soft shading (Lin et al., 2023a) to regularize the geometry. For regularization loss, we only use the orientation loss proposed by (Poole et al., 2023). We note that both of these two techniques only help to smooth the geometry and have little effect on the content in our case. We do not use any sparsity loss to force the separation between foreground the background but instead achieve it by replacing background with random colors. It is also worth noting that MVDream is theoretically combinable with other SDS variants such as SJC (Wang et al., 2023a) and VSD(Wang et al., 2023c), but this is out of the scope of this paper.

## 3.3 MULTI-VIEW DREAMBOOTH FOR 3D GENERATION

As shown in the right of Fig. (3), after training the multi-view diffusion model, we consider extending it to a DreamBooth model (Ruiz et al., 2023) for 3D DreamBooth applications. Thanks to the generalization of the multi-view diffusion model, we found its multi-view ability can be maintained after fine-tuning. Specifically, we consider two types of loss, an image fine-tuning loss and a parameter preservation loss. Formally, let $\mathcal{X}_{id}$ indicates the set of identity images, our loss for DreamBooth is

$$\mathcal{L}_{DB}(\theta, \mathcal{X}_{id}) = L_{LDM}(\mathcal{X}_{id}) + \lambda\frac{\|\theta - \theta_0\|_1}{N_\theta}, \tag{3}$$

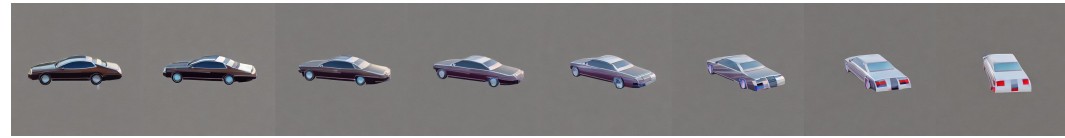

(a) Temporal Attention

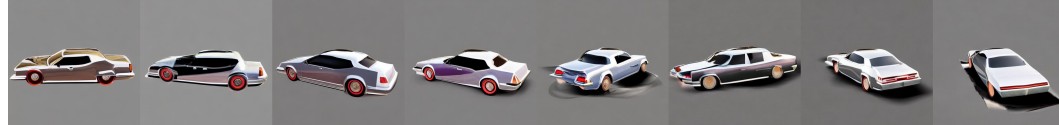

(b) Additional 3D Self-Attention module

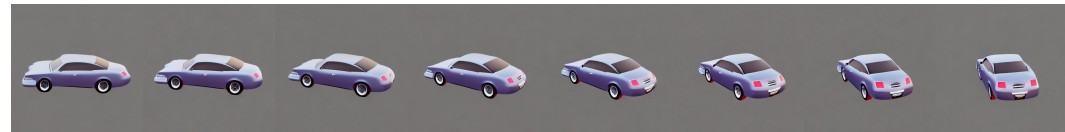

(c) Inflated 3D Self-Attention

Figure 4: Comparison between using different types of attention modules for learning multi-view consistency as discussed in Sec 3.1.1. Same random seed used for testing.

where $\mathcal{L}_{LDM}$ is the image diffusion loss (Rombach et al., 2022), $\theta_0$ is the initial parameter of original multi-view diffusion, $N_\theta$ is the number of parameters, and $\lambda$ is a balancing parameter set to 1. See Appendix for more training details.

To obtain a 3D model, we follow the process in Sec. 3.2 by replacing the diffusion model with the DreamBooth model. Note that original DreamBooth3D (Raj et al., 2023) used a three-stage optimization: partial DreamBooth, multi-view data generation, and multi-view DreamBooth. In comparison, our method capitalizes on the consistency of diffusion model and streamlines the process by training a multi-view (MV) DreamBooth model directly followed by 3D NeRF optimization.

## 4 EXPERIMENTS

To evaluate the multi-view diffusion models, we fine-tune the open-sourced stable diffusion 2.1 model (sta) on the Objaverse dataset (Deitke et al., 2023) and LAION dataset (Schuhmann et al., 2022) for experiments. We refer the readers to the Sec. A.6 for more training details. We evaluate our method on three tasks: (1) multi-view image generation for evaluating image quality and consistency (Sec. 4.1), (2) 3D (NeRF) generation with multi-view score distillation as a main downstream task (Sec. 4.2), and (3) DreamBooth for personalized 3D generation (Sec. 4.3).

### 4.1 MULTI-VIEW IMAGE GENERATION

We first compare the three choices of attention module for modeling cross-view consistency: (1) 1D temporal self-attention that is widely used in video diffusion models, (2) a new 3D self-attention module that is added onto existing model, and (3) inflating existing 2D self-attention module for 3D attention, as explained in 3.1.1. To show the difference between these modules, we trained the model with 8 frames across a 90 degree view change, which is closer to a video setting. We keep the image resolution at $512 \times 512$ as the original SD model in this experiment. As shown in Fig. (4), we found that even with such limited view angle change on static scenes, temporal self-attention still suffers from content drift. We hypothesize that this is because temporal attention can only interchange information between the same pixel from different frames while the corresponding pixels could be far away from each other in different views. Adding new 3D attention, on the other hand, leads to severe quality degradation without learning consistency. We believe this is because learning new parameters from scratch takes more training data and time, which is unsuitable for our case. The proposed strategy of inflated 2D self-attention achieves the best consistency among all without losing generation quality. We note that the difference between these modules would be smaller if we reduce the image size to 256 and the number of views to 4. However, to achieve the best consistency, we keep our choice for the following experiments based on our initial observations.

| Model | FID↓ | IS↑ | CLIP↑ |
|---|---|---|---|
| Validation set | N/A | $12.90 \pm 0.66$ | $30.12 \pm 3.15$ |
| Multi-view Diffusion (3D data only) | 40.38 | $12.33 \pm 0.63$ | $29.69 \pm 3.36$ |
| Multi-view Diffusion (3D + LAION 2D data) | 39.04 | $12.97 \pm 0.60$ | $30.38 \pm 3.50$ |

Table 1: Quantitative evaluation on image synthesis quality. DDIM sampler is used for testing.

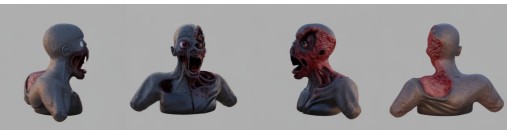

Zombie bust, terror, 123dsculpt, bust, zombie

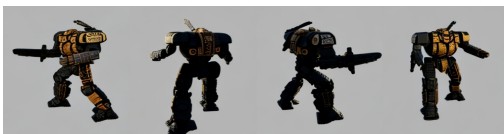

Battletech Zeus with a sword!, tabletop, miniature, battletech, miniatures, wargames, 3d asset

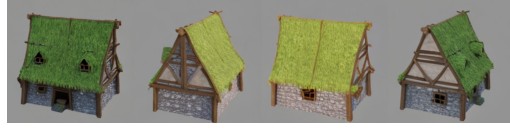

Medieval House, grass, medieval, vines, farm, middle-age, medieval-house, stone, house, home, wood, medieval-decor, 3d asset

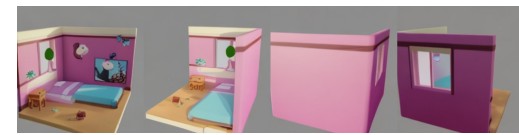

Isometric Slowpoke Themed Bedroom, fanart, pokemon, bedroom, assignment, isometric, pokemon3d, isometric-room, room-low-poly, 3d asset

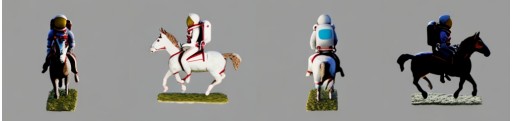

An astronaut riding a horse

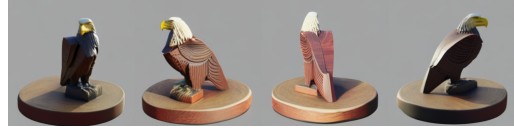

A bald eagle carved out of wood

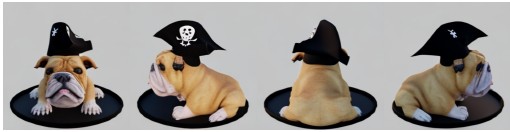

A bull dog wearing a black pirate hat

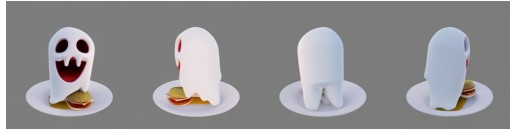

a DSLR photo of a ghost eating a hamburger

Figure 5: Example **images** generated by our model using training and testing prompts. The first two rows show the synthesized images from training prompts. The bottom four examples are images generated using unseen prompts. See Appendix for more results.

In Table 1, we conduct a quantitative comparison over generation quality and text-image consistency. We randomly choose 1,000 subjects from the held-out validation set and generate 4-view images using the given prompts and camera parameters. Frechet Inception Distance (**FID**) (Heusel et al., 2017) and Inception Score (**IS**) (Salimans et al., 2016) are calculated to measure the image quality while the CLIP score (Radford et al., 2021) is used to measure the text-image consistency, which is averaged across all views and subjects. Overall, our models can achieve a similar IS and CLIP score as the training set, indicating good image quality and text-image consistency. We found that fine-tuning with 3D data alone could lead to deteriorated image quality. On the other side, combining 3D data with a 2D dataset (LAION) for joint training mitigates the problem. Please also refer to Sec. A.4 and Fig. (12) for more qualitative examples for comparison.

In Fig. (5), we show a few qualitative examples from our model. It can be seen that our multi-view model can generate images from unseen prompts that are possibly counterfactual and in a different style from training prompts. Like training, we append the text with ", 3d asset" for generation.

## 4.2 3D GENERATION WITH MULTI-VIEW SCORE DISTILLATION

We compare our multi-view SDS with existing text-to-3D methods. We implement our method in threestudio (thr) and use reproduced baselines in this framework for comparison. We consider the following baselines: dreamfusion-IF (Poole et al., 2023), magic3d-IF (Lin et al., 2023a), text2mesh-IF (Tsalicoglou et al., 2023), and prolificdreamer (Wang et al., 2023c). Note that the implementation in

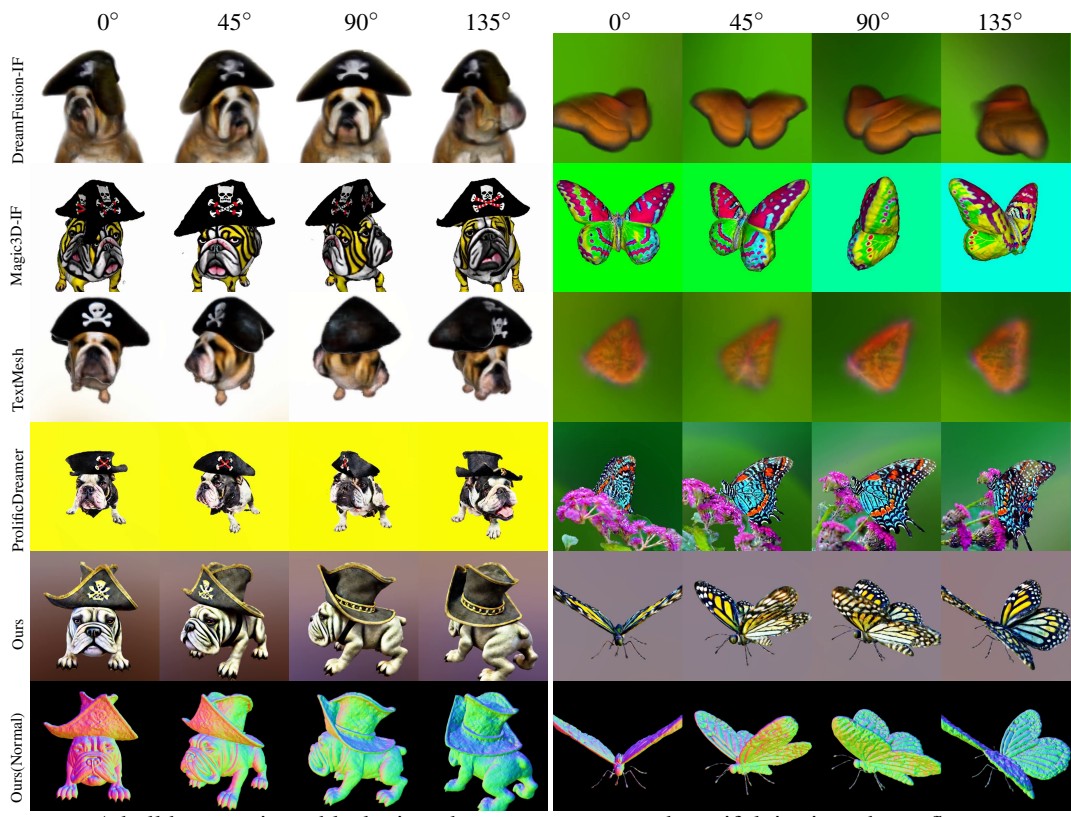

A bulldog wearing a black pirate hat      beautiful, intricate butterfly

Figure 6: Comparison of text-to-3D (**NeRF**) generation. See Appendix for more results.

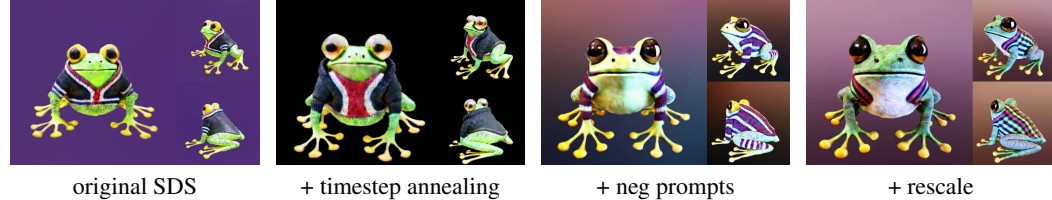

original SDS   + timestep annealing   + neg prompts   + rescale

Figure 7: Effects of different techniques that we apply in multi-view SDS. The input text is "a DSLR photo of a frog wearing a sweater".

threestudio could be different from the original papers. For example, dreamfusion-IF, magic3d-IF and text2mesh-IF use DeepFloyd (dee) as their guidance model while original ones use Imagen (Saharia et al., 2022) and eDiff-I (Balaji et al., 2022), which are private. Original dreamfusion uses Mip-NeRF 360 (Barron et al., 2022) for 3D representation, but all methods here (including ours) use multi-resolution hash-grid (Müller et al., 2022) (except Text2Mesh that uses SDF). We believe that these baselines represent the best re-implementation we could find. To test the performance comprehensively, we collected 40 prompts from various sources, including prior studies, internet prompts for image generation, and user inputs from a 3D generation website (lum). As shown in Fig. (6), all baselines suffer from severe multi-view inconsistency. Magic3D performs relatively better by using DMTet Shen et al. (2021) for second stage refinement. Prolificdreamer shows good texture quality for every view, but jointly they do not appear to be a reasonable 3D object. In comparison, the proposed method generates high quality 3D assets in a much more stable way. In terms of running time, dreamfusion-IF, text2mesh-IF and MVDream takes about 2 hours on a single V100 GPU. Magic3D-IF-SD takes about 3.5 hours and prolificdreamer takes more than 10 hours.

Fig. (7) shows an ablation study of the proposed techniques for score distillation introduced in Sec. 3.2. Adding time step annealing helps to make the shape more complete. Adding negative

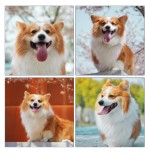 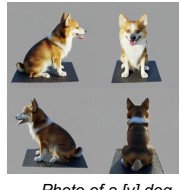 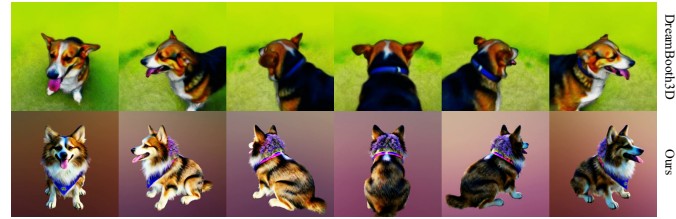

*Photo of a [v] dog*

Figure 9: Illustration of MVDreamBooth Results. From the inputs, we show the generated multi-view images given a description prompt at bottom. On the right, we show comparison of NeRF rendered results from ours with and DreamBooth3D (Raj et al., 2023). Notice ours perform better on the object details such as furry skin. See Appendix for more results.

prompts significantly improves the visual style. Adding the CFG rescale further makes the texture color more natural.

To further validate the stability and quality of our model, we conducted a user study on the generated models from 40 prompts. Each user is given 5 rendered videos and their corresponding text input and is asked to select a preferred 3D model among all. 914 feedbacks from 38 users were collected and the result is shown in Fig. (8). On average, 78% users prefer our model over others. That is, our model is preferred over the best of all baselines in most cases. We believe this is a strong proof of the robustness and quality of the proposed method. Please see the supplementary materials for more visual results.

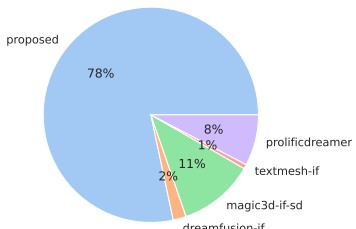

Figure 8: User study.

### 4.3 Multi-View DreamBooth.

In Fig. (9), we compare 3D models generated from MVDream and DreamBooth3D (Raj et al., 2023). We found that our results have higher quality with better object details such as the curly hair and fur texture on the dog. This is because during the NeRF training process with SDS loss, our MV DreamBooth diffusion models produce higher geometry consistency. We provide additional results in our project page and supplementary materials.

## 5 Discussion and Conclusion

**Conclusion**   In this paper, we present the first multi-view diffusion model that is able to generate a set of multi-view images of an object from any given text. By fine-tuning a pre-trained text-to-image diffusion model on a mixture of 3D rendered data and large scale text-to-image data, our model is able to maintain the generalizability of the base model while achieving multi-view consistent generation. We show that the multi-view diffusion model can serve as a good 3D prior and can be applied to 3D generation via SDS, which leads to better stability and quality than current open-sourced 2D lifting methods. Finally, the multi-view diffusion model can also be trained in a few-shot setting for personalized 3D generation (multi-view DreamBooth).

**Limitation**   We observe the following limitations of our current multi-view diffusion model. For the moment, the model can only generate images at a resolution of $256 \times 256$, which is smaller than the $512 \times 512$ of the original stable diffusion. Also, the generalizability of our current model seems to be limited by the base model itself. For both aforementioned problems, we expect them to be solved by increasing the dataset size and replacing the base model with a larger diffusion model, such as SDXL (SDX). Furthermore, we do observe that the generated styles (lighting, texture) of our model are affected by the rendered dataset. Although it can be alleviated by adding more style text prompts, it also indicates that a more diverse and realistic rendering is necessary to learn a better multi-view diffusion model, which could be costly.

## 6 ETHICS STATEMENT

The multi-view diffusion model proposed in this paper aims to facilitate the 3D generation task that is widely demanded in gaming and media industry. We do note that it could be potentially applied to unexpected scenarios such as generating violent and sexual content by third-party fine-tuning. Built upon the Stable Diffusion model (Rombach et al., 2022), it might also inherit the biases and limitations to generate unwanted results. Therefore, we believe that the images or models synthesized using our approach should be carefully examined and be presented as synthetic. Such generative models may also have the potential to displace creative workers via automation. That being said, these tools may also enable growth and improve accessibility for the creative industry.

## 7 REPRODUCIBILITY STATEMENT

Our MVDream model is built from publicly available models and datasets, such as Stable Diffusion (Rombach et al., 2022), LAION (Schuhmann et al., 2022) and Objaverse (Deitke et al., 2023). With all the implementation details provided, we believe it should be straightforward to re-produce the algorithm. Besides, we will release our code as well as model checkpoints publicly after the paper submission.

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

# A   APPENDIX

## A.1   COMPARISON WITH IMAGE-TO-3D METHOD

In Fig. (10), we compare our results with an state-of-the-art image-to-3D method, namely Zero123-XL (Liu et al., 2023). In particular, we first generate an image using the same prompt with SDXL (Podell et al., 2023), segment the object to remove background, and then provide the image as input to the image-to-3D system using Zero123-XL in threestudio. We notice two drawbacks of such a pipeline compared to our text-to-multi-view diffusion model:

1. Zero123-XL is trained on 3D datasets only, whose objects are mostly aligned with simple poses (e.g. T-pose for characters). So the image-to-3D results tend to be distorted if the input image has complicated poses and view angles that are different from 3D dataset. Specifically, when the object is geometrically spanned in the direction of the depth, the generated 3D tends to be too flat. This can be seen from the examples of eagle.

2. Zero123-XL can hardly generate geometrically consistent and richly textured novel-view images, and therefore the score distilled objects tend to have less details, yielding a smooth, blurred and relatively uniform appearance on the other sides different from the input view, as shown with the example of bulldog.

## A.2   EFFECT OF NUMBER OF VIEWS

In Fig. (11), we compare the SDS with diffusion models trained with different number of views. All the models are trained with the same settings except the number of views. Camera embeddings is used for all models. We found that the 1-view model, in spite of its camera-awareness, still suffer

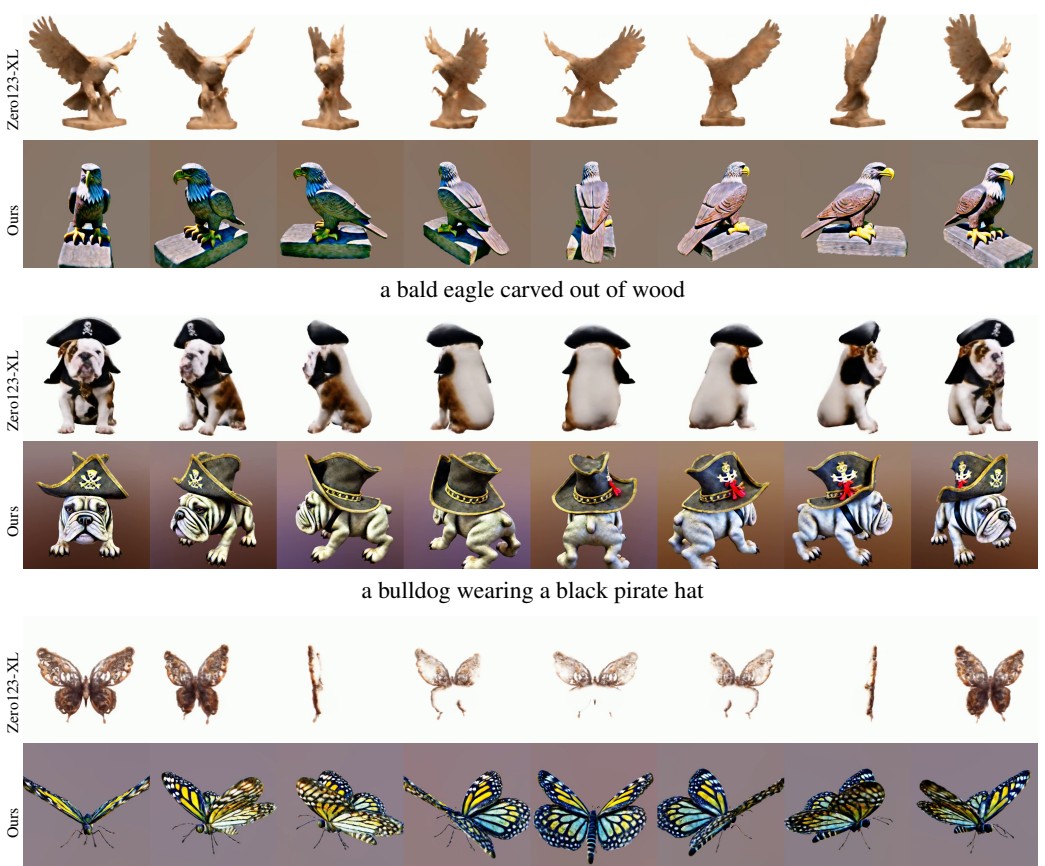

a bald eagle carved out of wood

a bulldog wearing a black pirate hat

beautiful, intricate butterfly

Figure 10: Comparison with text-to-image-to-3D using SDXL + Zero123XL.

1 view          2 view          4 view (proposed)

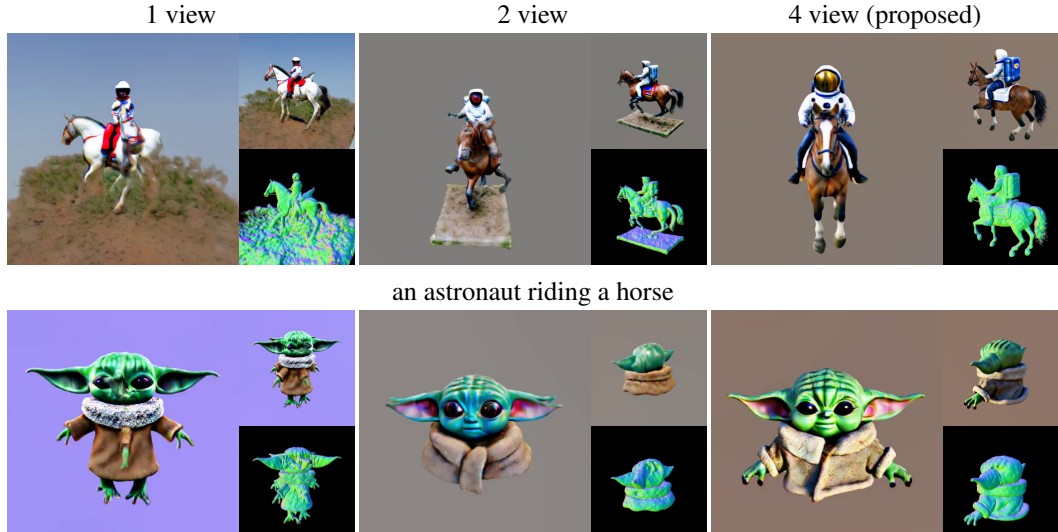

an astronaut riding a horse

baby yoda in the style of Mormookiee

Figure 11: Comparison between SDS with diffusion models trained with different number of views. All models have camera embeddings as inputs.

3D data only                  3D + 2D (LAION) data

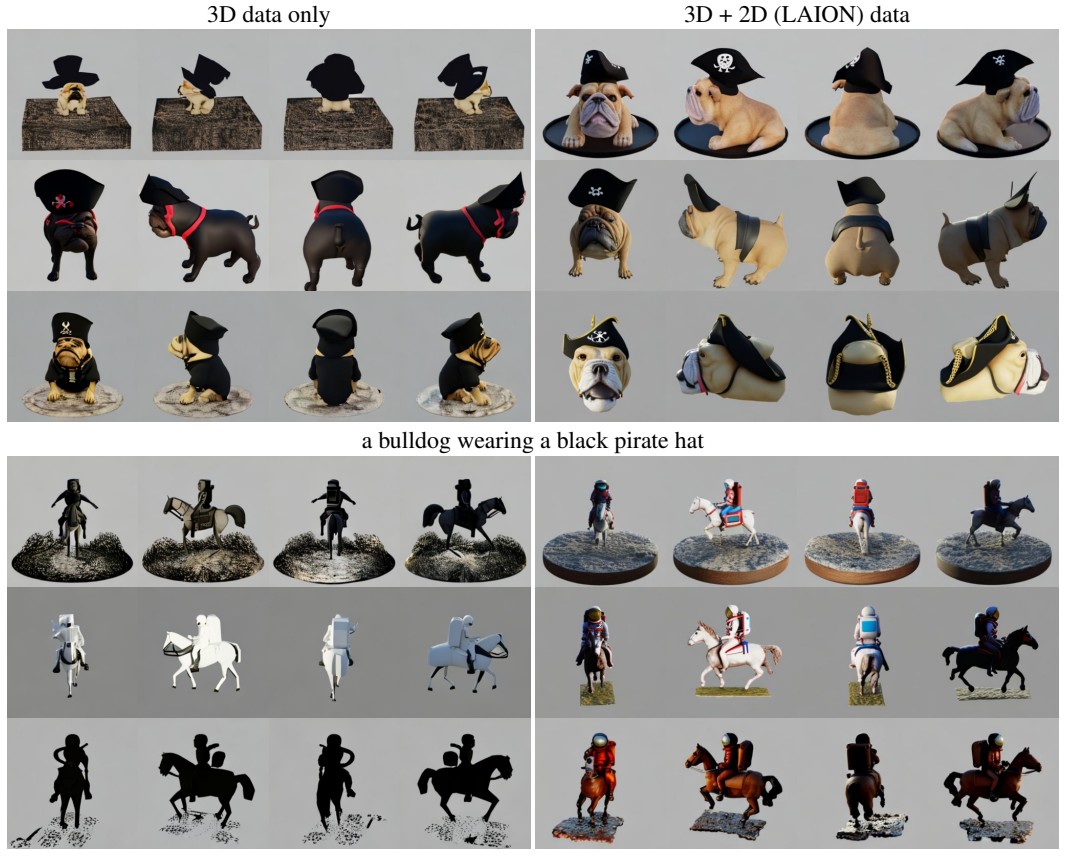

a bulldog wearing a black pirate hat

an astronaut riding a horse

Figure 12: Qualitative comparison between different multi-view diffusion models in Table 1. The same random seed is used for each row for comparison.

from a severe Janus problem, partially due to imperfect camera annotations. The 2-view model greatly reduces the multi-face problem, but it still could happen to some extent. Finally, the 4-view model barely suffers from any multi-view consistency issues.

## A.3 TRAINING WITH 2D DATA

Fig. (12) shows the qualitative comparison between training with and without 2D data for multi-view diffusion model. Empirically, we found that adding the 2D data has a clear improvement for the model generalizability, leading to better image quality and text-image correspondence.

## A.4 TRAINING WITH RANDOM CAMERA

In the main paper, our model is trained for 4 orthogonal views with the same elevation angle. Thus, a question arises: can the method generalize to scenarios involving random camera views rather than orthogonal ones? We conduct such an experiment and found that the proposed 3D self-attention network can indeed generate random multi-view images if trained on such data. In spite of worse quality compared to the orthogonal model, an interesting finding is that such a random-view model able to generate a much larger number of views during inference, as shown in Fig. (13). In our experiments, we found it able to generate 64 views and potentially more, even though it is trained on 4 views.

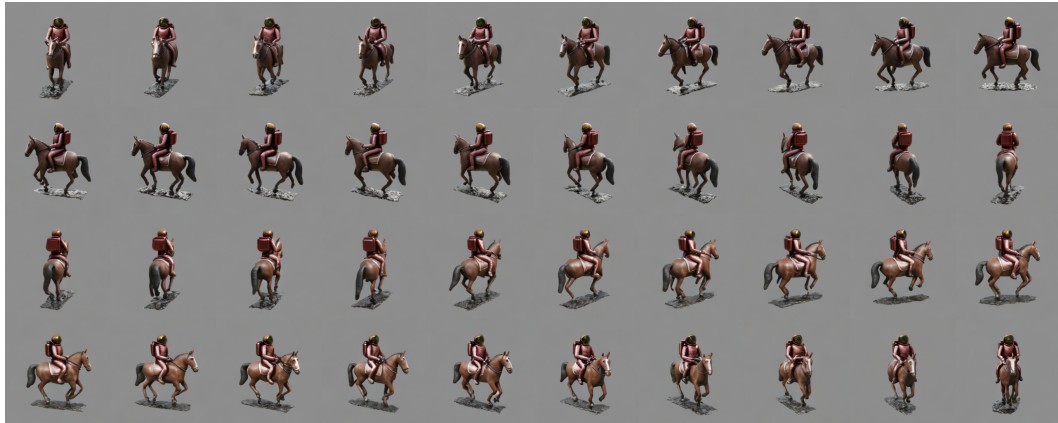

Figure 13: Multi-view diffusion model trained on random views is able to generate more a larger number of views than training (40 views for example here).

## A.5 $\mathbf{x}_0$ RECONSTRUCTION LOSS FOR SDS

The original SDS loss does not have an explicit form, instead it is defined by its gradient:

$$\nabla_\phi \mathcal{L}_{SDS}(\theta, \mathbf{x} = g(\phi)) = \mathbb{E}_{t,c,\epsilon}\left[w(t)(\epsilon_\theta(\mathbf{x}_t; y, \mathbf{c}, t) - \epsilon)\frac{\partial \mathbf{x}}{\partial \phi}\right] \quad (4)$$

where $\mathbf{c}$ is the additional camera condition for our specific model, $\mathbf{x}_t = \alpha_t \mathbf{x} + \sigma_t \epsilon$ is the sampled noisy image, and $w(t)$ is a hyper-parameter that controls the loss weight at different time steps. $\alpha_t$ and $\sigma_t$ are the signal and noise scale controlled by the noise schedule, where $\alpha_t^2 + \sigma_t^2 = 1$. Let us denote $\epsilon_\theta(\mathbf{x}_t; y, \mathbf{c}, t)$ as $\epsilon_\theta$ for simplicity. Given estimated $\epsilon_\theta$ predicted by the model, we can reversely estimate $\hat{\mathbf{x}}_0$ by:

$$\begin{aligned}
\hat{\mathbf{x}}_0 &= \frac{\mathbf{x}_t - \sigma_t \epsilon_\theta}{\alpha_t} \\
&= \frac{\alpha_t \mathbf{x} + \sigma_t \epsilon - \sigma_t \epsilon_\theta}{\alpha_t} \\
&= \mathbf{x} + \frac{\sigma_t}{\alpha_t}(\epsilon - \epsilon_\theta)
\end{aligned} \quad (5)$$

**Algorithm 1:** Pseudocode for MVDream training

---

**Data:** $\mathcal{X}, \mathcal{X}_{mv}$
**for** $i \leftarrow 1$ *to* $n - 1$ **do**
    sample $mode \sim U(0, 1)$;
    **if** $mode \leq 0.7$ **then**
        select a random 3D sample from $\mathcal{X}_{mv}$;
        $\mathbf{x} \leftarrow$ 4 random orthogonal views out of 32 views;
        $\mathbf{c} \leftarrow$ camera extrinsics;
    **else**
        $\mathbf{x} \leftarrow$ 4 random images from $\mathcal{X}$;
        $\mathbf{c} \leftarrow \varnothing$ ;
    $y \leftarrow$ text descriptions;
    sample $t \sim U(0, 1000)$;
    $\mathbf{x}_t \leftarrow$ add_noise($\mathbf{x}, t$);
    forward_and_backward($\theta, \mathbf{x}, \mathbf{x}_t, y, \mathbf{c}, t$)

---

So for the reconstruction loss $\mathcal{L}_{SDS} = \mathbb{E}_{t,\mathbf{c},\epsilon}\left[\|\hat{\mathbf{x}}_0 - \mathbf{x}\|_2^2\right]$, if we ignore $\frac{\partial \hat{\mathbf{x}}_0}{\partial \phi}$ like in SDS, its gradient is given by:

$$
\begin{aligned}
\nabla_\phi \mathcal{L}_{SDS}(\theta, \mathbf{x} = g(\phi)) &= \mathbb{E}_{t,\mathbf{c},\epsilon}\left[2(\mathbf{x} - \hat{\mathbf{x}}_0)\frac{\partial \mathbf{x}}{\partial \phi}\right] \\
&= \mathbb{E}_{t,\mathbf{c},\epsilon}\left[\frac{2\sigma_t}{\alpha_t}(\epsilon_\theta - \epsilon)\frac{\partial \mathbf{x}}{\partial \phi}\right],
\end{aligned}
\tag{6}
$$

which is equivalent to the original SDS with $w(t) = \frac{2\sigma_t}{\alpha_t}$. Thus, we can further adjust the $\hat{\mathbf{x}}_0$ here with other tricks such as dynamic threshold (Saharia et al., 2022) or CFG rescale (Lin et al., 2023b).

## A.6 IMPLEMENTATION DETAILS

### A.6.1 DATA PREPARATION AND DIFFUSION MODEL TRAINING

We use the public Objaverse dataset (Deitke et al., 2023) as our 3D rendered dataset, which was the largest 3D dataset available by the time of this project. We simply use the names and tags of the objects as their text descriptions. Since this dataset is rather noisy, we filter the dataset with a CLIP score to remove the objects whose rendered images are not relevant to its name, which leads to about 350K objects at the end. For each object, we first normalize the object to be at the center within a bounding box between $[-0.5, 0.5]$, and then choose a random camera fov between $[15, 60]$ and elevation between $[0, 30]$. The camera distance is chosen as the object size ($0.5$) times NDC focal length with a random scaling factor between $[0.9, 1.1]$. A random HDRI from blender is used for lighting. 32 uniformly distributed azimuth angles are used for rendering, starting from 0 degree. To increase the number of training examples, we render each object twice with different random settings. The rendered images are saved in RGBA format where the background is filled with a random grayscale color during training.

During the training, we sample a data batch with a 70% chance from the 3D dataset and 30% chance from a subset of LAION dataset (Schuhmann et al., 2022). For the 3D dataset, 4 random views which are orthogonal to each other are chosen from the 32 views for training. The $4 \times 4$ camera parameters are normalized onto the sphere for inputs, i.e. it only represents the rotation. We fine-tune our model from the Stable Diffusion v2.1 base model ($512 \times 512$ resolution) (sta), where we keep their settings for the optimizer and $\epsilon$-prediction. We use a reduced image size of $256 \times 256$ and a total batch size of 1,024 (4,096 images) for training and fine-tune the model for 50,000 steps. The training takes about 3 days on 32 Nvidia Tesla A100 GPUs. To compensate for the visual style difference between the Objaverse and LAION dataset, we append the text ", 3d asset" to the 3D data if the keyword "3d" is not in the prompt. For DreamBooth fine-tuning, we train the model around for 600 steps with a learning-rate as $2e - 6$, weight decay as 0.01 and a batch size of 4. A pseudo-code for training is provided in Algorithm. 1.

### A.6.2 SDS Optimization

For multi-view SDS, we implement our multi-view diffusion guidance in the threestudio (thr) library, which has implemented most state-of-the-art methods for text-to-3D generation under a unified framework. We use the implicit-volume implementation in threestudio as our 3D representation, which includes a multi-resolution hash-grid and a MLP to predict density and RGB. For camera views, we sample the camera in the same way as how we render the 3D dataset. See Sec. A.6.1 for more details. The 3D model is optimized for 10,000 steps with an AdamW optimizer (Kingma & Ba, 2014) at a learning rate of 0.01. For SDS, the maximum and minimum time steps are decreased from 0.98 to 0.5 and 0.02, respectively, over the first 8,000 steps. We use a rescale factor of 0.5 for the CFG rescale trick. The rendering resolution starts at 64×64 and is increased to 256×256 after the 5,000 steps. We also turn on soft shading as in (Lin et al., 2023a) after 5,000 steps. The background is replaced with 50% chance. The SDS process takes about 1.5 hour on a Tesla V100 GPU with shading and 1 hour without shading.

### A.7 Additional Results

In Fig. (14), we show additonal image generation results from our model using the prompts from the training dataset. In Fig. (15) and (16), we show additional comparison results of 3D generation using our method and baseline. In Fir. (17) we show additional results of DreamBooth using our method and DreamBooth3D (Raj et al., 2023).

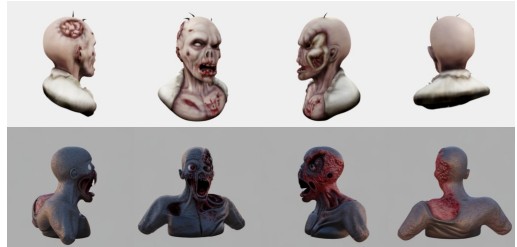

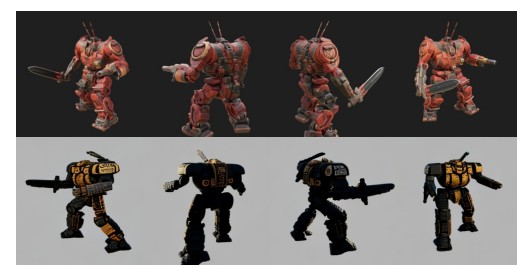

Zombie bust, terror, 123dsculpt, bust, zombie

Battletech Zeus with a sword!, tabletop, miniature, battletech, miniatures, wargames

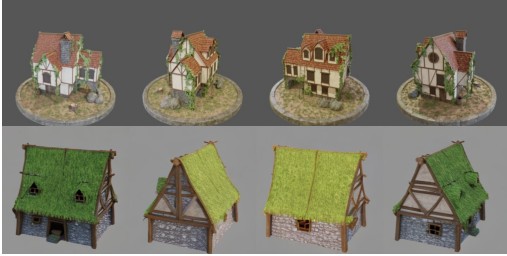

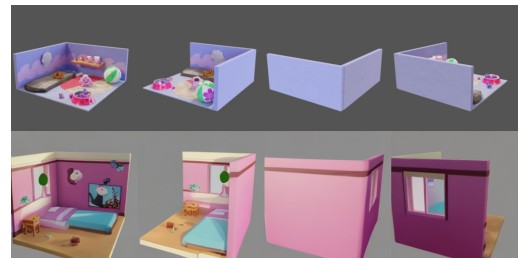

Medieval House, grass, medieval, vines, farm, middle-age, medieval-house, stone, house, home, wood, medieval-decor

Isometric Slowpoke Themed Bedroom, fanart, pokemon, bedroom, assignment, isometric, pokemon3d, isometric-room, room-low-poly

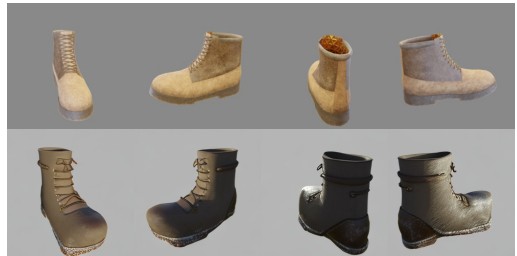

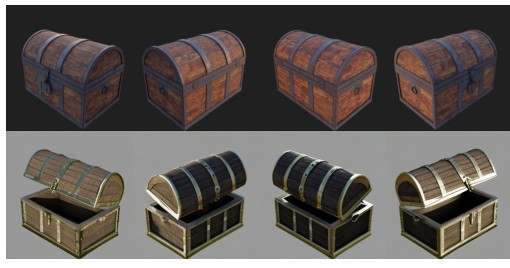

Old Boot (Left), cloth, boot, old, substancepainter, substance, character

Old Treasure Chest, chest, vintage, treasure, old, substancepainter, substance

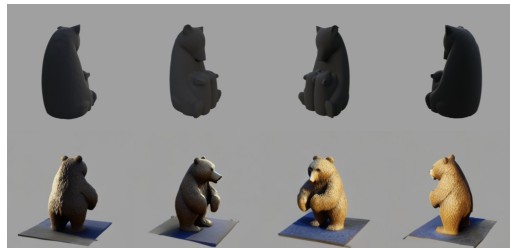

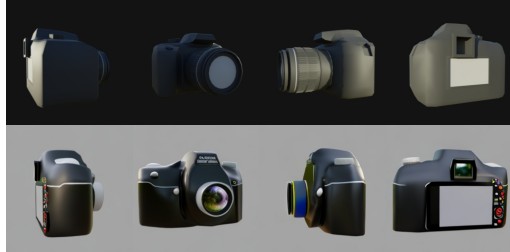

UCSF's Bear Mascot Sculpture, bear, library, sanfrancisco, photogramm, ucsf, kalmanovitz, bufano

DSLR Camera, photography, dslr, camera, noobie, box-modeling, maya

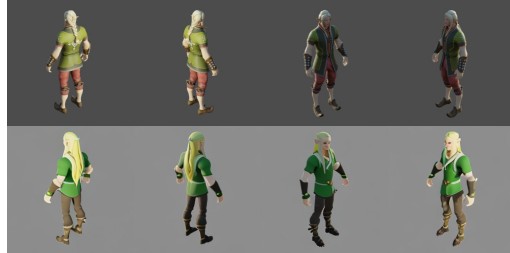

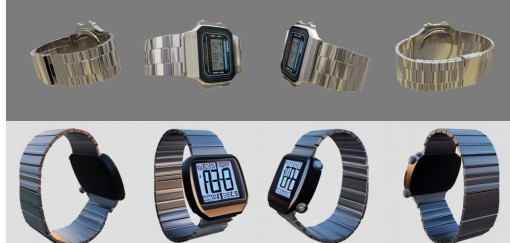

"Legolas" Custom Outfit, custom, free-model, fortnite-

Digital Wristwatch, wristwatch, timepiece, digital, watch

Figure 14: Example images generated by our multiview diffusion model using prompts in training set. In each example, the top and bottom row are the real and synthesized images, respectively.

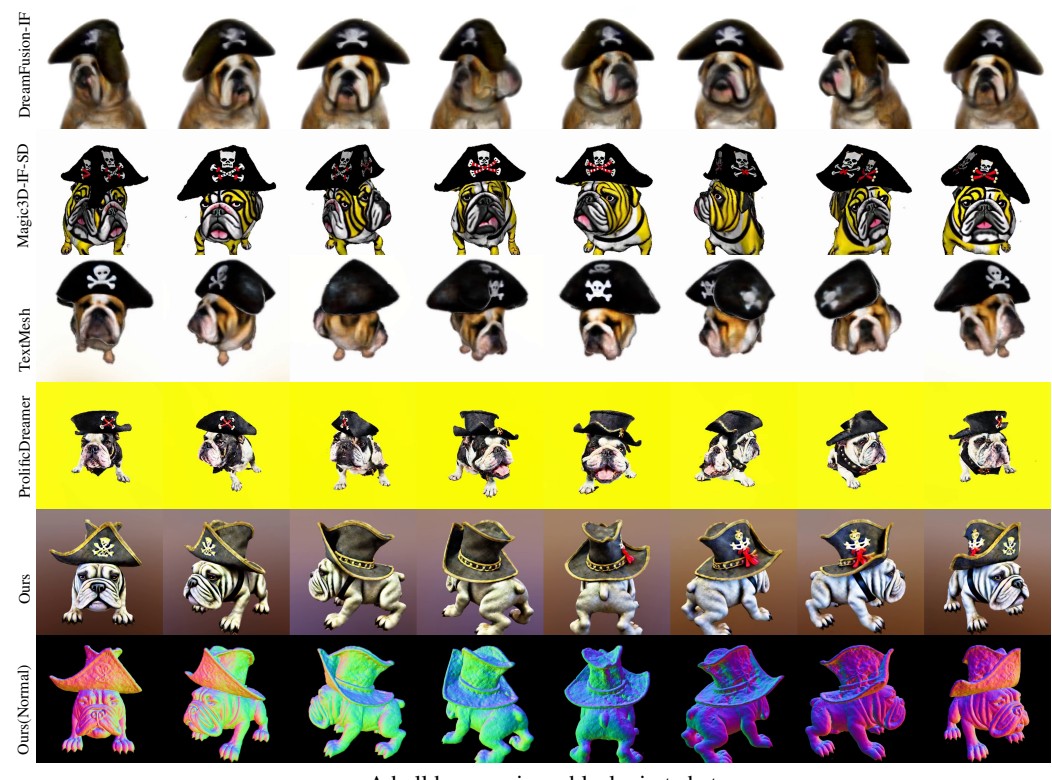

A bulldog wearing a black pirate hat

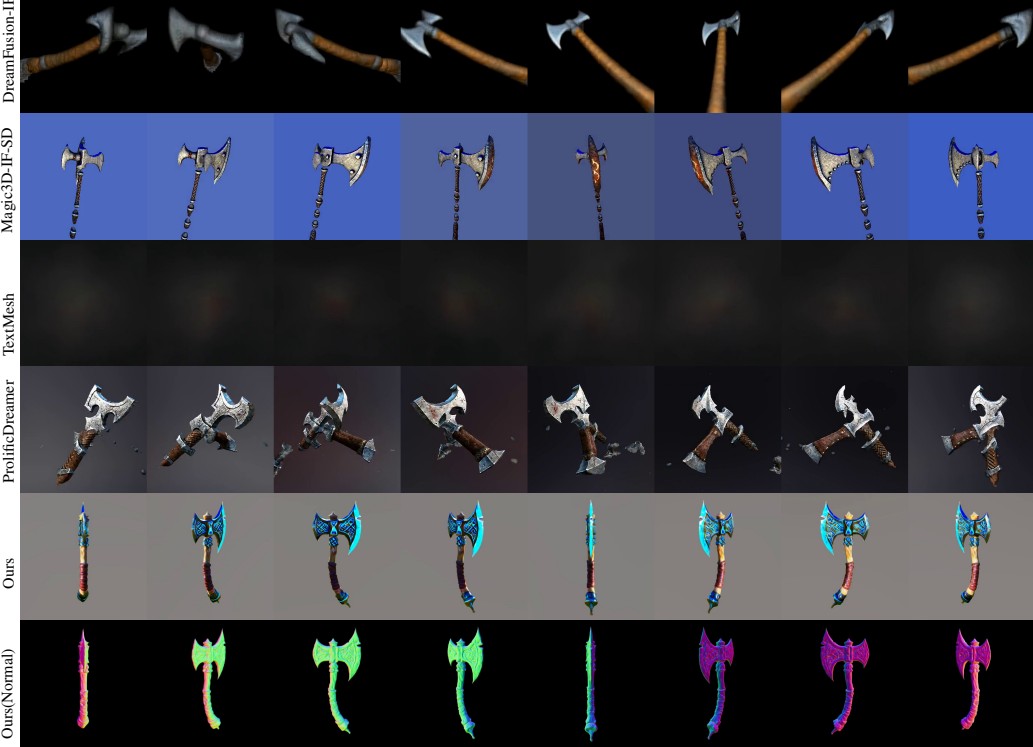

Viking axe, fantasy, weapon, blender, 8k, HD

Figure 15: Comparison of 3D generation between baselines and our method.

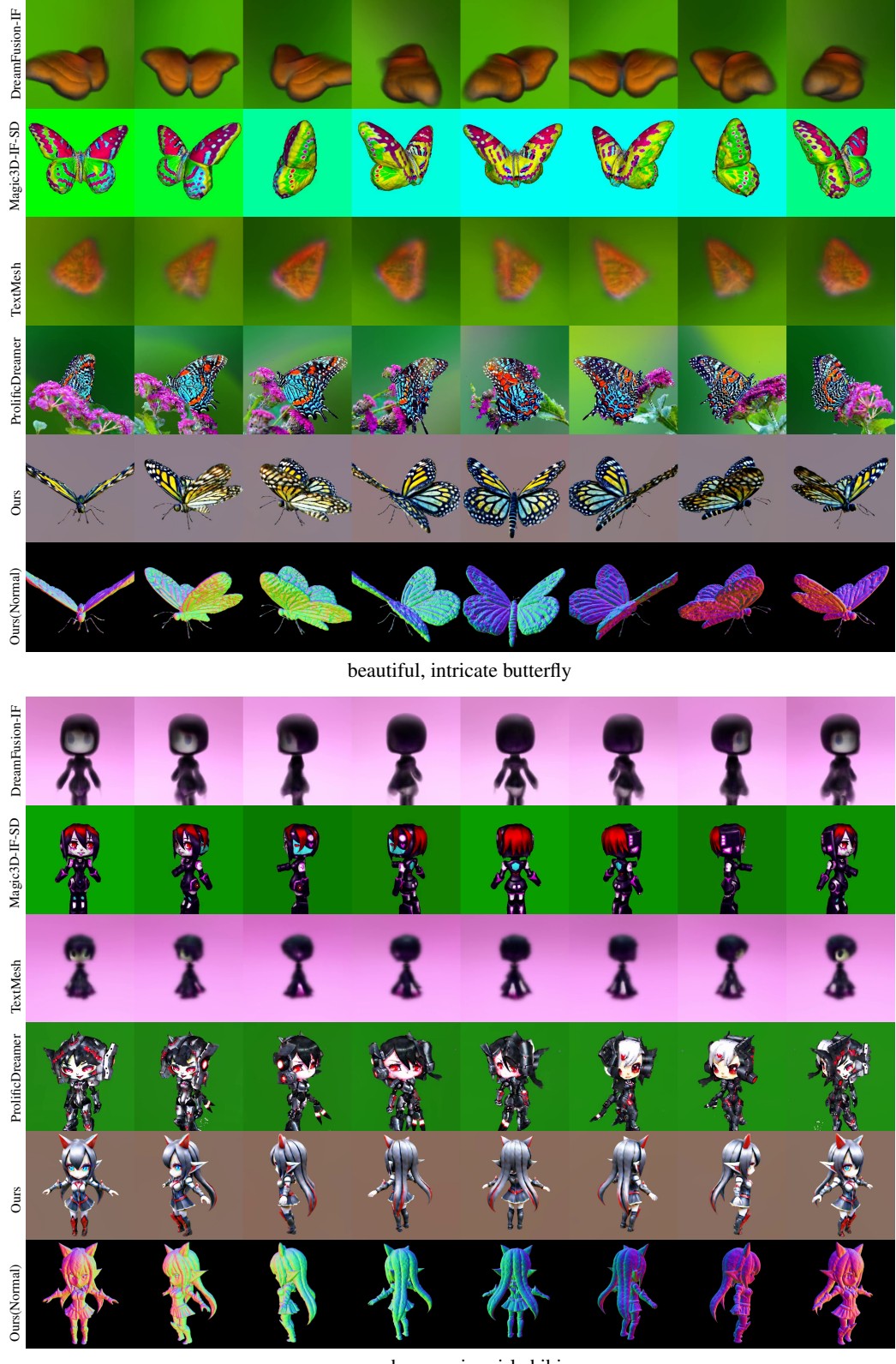

beautiful, intricate butterfly

mecha vampire girl chibi

Figure 16: Comparison of 3D generation between baselines and our method.

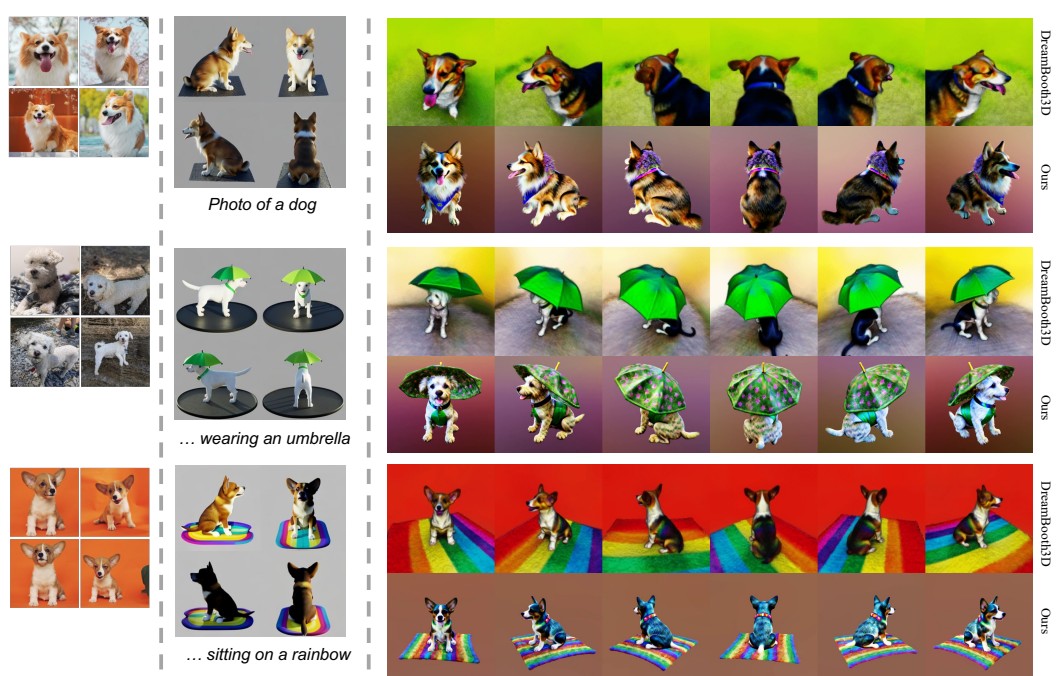

Figure 17: MVDream DreamBooth Results.

