# OpenReview forum: "MVDream: Multi-view Diffusion for 3D Generation"
_ICLR.cc/2024/Conference — ICLR 2024 poster_

### Official Review · Reviewer_HRvB · 2023-10-29

**Soundness:** 3 good
**Presentation:** 4 excellent
**Contribution:** 3 good
**Rating:** 8
**Confidence:** 4

**Summary:**

The paper proposes a novel method for performing text-to-3D generation using the score-distillation loss with a 2D diffusion model. The key insight is that using a diffusion model which is fine-tuned with multi-view data, and thus has some notion of view consistency, is able to provide better gradients in optimization and result in a better 3D representation consistent with the text prompt, avoiding classic text-to-3D artifacts such as the multi-face or Janus problem. The paper trains the score distillation diffusion model with multi-view data by augmenting it to produce 4 multi-view images of an object rather than a single image and uses rendered images from synthetic data as supervision. Using this method, the paper then demonstrates that using this diffusion model in score distillation enables higher quality generations, and importantly, removes the Janus problem as the gradients from each score-distillation rendered image are not biased towards the canonical "front" view of an object.

**Strengths:**

In my opinion, the strengths of the paper are:
1. The paper is extremely clear, and the method is justified intuitively. I found the discussion on using the 2D diffusion model to generate images and then using these images directly to train a 3D representation to be very insightful, as this is a common baseline that naturally follows from the contribution of making the diffusion model partially 3D-aware.
2. The generated results are extensive, and qualitatively, the quality of the generations is very high and it seems like the results avoid the Janus problem.
3. The evaluations are comprehensive, using multiple sets of metrics, ablating potential baselines, and including a user study.

**Weaknesses:**

In my opinion, the weaknesses of the paper are:
1. One potential major weakness is that the robustness of the method does not seem to be tested. In terms of the generated it results, it would be good to see multiple versions generated from the same prompt (i.e., corresponding to something like variance in quantitative metrics from various trained models). This is very important because text-to-3D is notoriously brittle, and it's important to understand if the proposed contribution removes the Janus problem 5% of the time, 50% of the time, or 100% of the time. Without this sort of comparison and extensive evaluation, it's impossible to judge the magnitude of the contribution.
2. A minor weakness is that there seem to be a large number of hyperparameters where it's not entirely clear why their values were chosen. Why use a multi-view diffusion model outputting only 4 views instead of more? The justification for how camera parameters are embedded also seems a bit ad-hoc. I don't see this as a major weakness as they lead to good performance, but I wonder about how brittle the method is as a result.

**Questions:**

1. Is the view dependent prompting used in DreamFusion completely removed in favor of using the camera position embedding in the multi-view diffusion model? Or is there still some notion of directionality included in the text prompt?

Overall, I think that the paper is meriting of acceptance. The description of the method is clear, and the proposed contribution makes intuitive sense and seems to resolve the Janus problem and increase the quality of the text-to-3D representations. The evaluation is very thorough, compares to other state-of-the-art methods in this field with a number of qualitative results and metrics, including a user study. I think that deeply studying the robustness of the method, and showing that the proposed contribution fixes the Janus problem 100% of the time consistently would elevate the paper to be significantly stronger, and I'd be happy to increase my review score in that case.

**Update after the author response**

Thank you for the additional clarifications. I do not have any more questions and choose to keep my score: weaknesses regarding hyperparameters and robustness have been addressed.

---

> ### Author Response · Authors · 2023-11-13
> **Response to Reviewer HRvB**
>
> We thank the reviewer for the detailed and helpful comments. The belows are our responses to reviewer's concerns:
>
> 1. **Janus Problem**\
> In the anonymous project page https://mvdream-anonymous.github.io/test_0.html, we have shown the qualitative comparison over 40 prompts without cherry picking nor prompt-wise parameter tuning. It could be seen that our model almost eliminates the multi-face Janus problem by a 100% chance (with a fixed configuration). We have also conducted an experiment for the same prompt but different random seeds. However, we found the difference to be rather subtle because of the large CFG guidance used during SDS: https://github.com/mvdream-anonymous/mvdream-anonymous.github.io/blob/master/static/seeds.mp4
>
> 2. **Hyper-parameters**
> - **number of views** Since we are proposing a new type of neural networks, it will inevitably comes with a few new hyper-parameters. For the number of views, as shown in Figure 11 in Appendix A.2, we found 4 views to a minimum number of views to ensure multi-view consistency for SDS. Therefore, we choose to stop at 4 views to balance the tradeoff between performance and resource cost.
> - **camera parameters** In our experiments, we have tried different types of camera inputs (16-dim extrinsics and 25-dim intrinsics+extrinsics) and different ways to embed them (cross attention vs. time embedding). All of these choices work to some extent. So we choose our current design due to its simpleness and empirical robustness.
> - Overall, our pipeline already has much less hyper-parameters compared with prior studies such as Dreamfusion and Magic3D. In particular, our model can in fact use SDS loss alone to generate 3D content while prior studies have many regularization losses to ensure 3D quality. Similar to the response to Question 1, we believe that the stableness of our method can be evaluated from the visual examples in the project page.
>
> 3. **View-dependent Prompts** \
> Yes, the view dependent prompting are completely removed in MVDream. We only give the original text descriptions, e.g. "an astronaut riding a horse", to the text encoder.

---

### Official Review · Reviewer_XNxK · 2023-10-29

**Soundness:** 2 fair
**Presentation:** 2 fair
**Contribution:** 4 excellent
**Rating:** 6
**Confidence:** 4

**Summary:**

This paper approaches the task of text-conditioned 3D object generation. It addresses the issues of prior work, namely the multi-face Janus problem and content drift problem; while adding DreamBooth-style results. The method supervises SDS against multiple views, which are rendered from a video diffusion-inspired text-to-multi-view model, which is trained upon large scale 3D-text data. The results are significantly better than prior work, but the experiments are difficult to follow.

**Strengths:**

The method shows a distinctive improvement over the state of the art.
- It clearly reduces the multi-face janus problem. In several shown examples, it eliminates this problem.
- It is selected as better than previous work at an extremely high rate: 78% vs. 4 other methods, combined.
- Objects are clearly consistent across 360degree views, and do not suffer from content drift
- Exhaustive results upon the project webpage make these conclusions clear

Utilizing 3D assets to train a text-to-3D model for score distillation is a creative and effective strategy. The paper even shows impressive DreamBooth-style results

**Weaknesses:**

Summary: the qualitative performance of the methods in this paper are impressive, but the experimental presentation in writing is poor. I think many changes are needed, but the results are quite impressive so I am still positive about this paper.

Experimental presentation is poor
- What is the point of each experiment? There is no overview section to clarify this. After multiple reads, I think I can infer, but reading was challenging.
- It is often very hard to figure out the point of an experiment or the details. For instance, upon first read, I was unclear at why Sec 4.1 was before 4.2, and if the main contribution of the paper was 4.1 and the distinction with 4.2.
- It is interesting to use a text-to-multi-view model to train a text-to-3D model. How well would the multi-view diffusion model work to generate 3D directly?
- There are generally lots of experimental details that need to be expanded. For instance, the attention module choice experiment is difficult to understand. Section 4.1 has the overview to the experiment and results in the same paragraph. The experiment isn’t fully specified, missing details such as what the caption is for Figure 4, or details of e.g. the 3D attention. The extent of this experiment is also one qualitative example, so it is hard to draw any confident conclusion.
- Why is the train set used for evaluation in Table 1, as opposed to a held-out set? What is the significance of evaluating a change in batch size? The results don’t change significantly, and increasing batch size isn’t a technical contribution. It was also hard to understand what data was used until after more than one read of the paper. The following was also confusing: “Adding text-to-image dataset (LAION) further improves the CLIP score and IS.” Is the difference between “no 2D data” and “proposed” training on LAION? This isn’t clarified anywhere. Why is this a contribution? Metrics change very little.
- Qualitative results in Figure 5 is a part of the same paragraph as the experiment in Table 4. Again, why is the training set shown? Is this the text-to-multi-view model or the text-to-3D model?
- What is the significance of adding negative prompts, or CFG rescaling? The reasoning for these isn’t clarified anywhere
- Dreambooth results are touched on briefly and seem disconnected from the rest of the paper. Why is this connected? Why does it work?

**Questions:**

See weaknesses

---

> ### Author Response · Authors · 2023-11-13
> **Response to Reviewer XNxK**
>
> We thank the reviewer for the careful and helpful comments. Here are our responses to the reviewer's concerns:
>
> 1. **point of experiments** \
> we have added an overview paragraph in the revised paper to explain the experiments. In particular, we evaluated our method on three tasks: (1) multi-view image generation for evaluating image quality and consistency (Sec. 4.1), (2) 3D (NeRF) generation with multi-view score distillation as a main downstream task (Sec. 4.2), and (3) DreamBooth for personalized 3D generation (Sec. 4.3).
> 2. **Sec 4.1 and 4.2** \
> Sec 4.1 (multi-view image generation) Sec 4.2 (multi-view score distillation) are both important contributions of our work. Since our model is essentially an image generation model, we put Section 4.1 before 4.2. We will explain this in the overview paragraph.
> 3. **Direct 3D Generation**\
> In order to generate 3D content directly without score distillation (optimization), a few-shot reconstruction model will be needed to generate 3D content from 4 views. However, we do not have access to such a few-shot reconstruction model for the moment. We believe this is achievable by training such a model or extending multi-view diffusion to more views. But this is out of the scope of this paper and will be part of the future work.
> 3. **Experiment Details** \
> We will try to add more explanations to the experiments, including the attention part. The main motivation of the attention experiment is to compare the multi-view consistency, for which we are not aware of any good quantitative metrics for the moment. So we choose to present qualitative results instead.
>
> 4. **Table 1**\
> We have updated the results on a held-out validation set.
> - We have removed the batch size comparison part and there are now two models trained with the same batch size.
> - Yes, "2D data" refers to the LAION dataset. We apologize for the confusion and will make it more clear. Figure 12 in Appendix shows the results between using 3D data alone and 2D+3D data. We observe a clear improvement in image quality by combining 2D and 3D data for training.
> 5. **Figure 5** \
> These are the images rather than the 3D models generated by our model. Here, we show both in-domain data (training) and out-of-domain examples (conterfactual text descriptions). We will explain this Figure more clearly.
> 6. **negative prompts and CFG rescaling**\
> The motivation of negative prompts and CFG rescaling were mentioned Sec 3.3. We will add a reference in the experiment back to this section.
> 7. **Dreambooth**\
> Dreambooth is an extended experiment of our multi-view diffusion model to evaluate its generalizability. It shows the capability how a multi-view diffusion model connects the knowledge from 2D and 3D domain. The explanation of why DreamBooth would work is a tricky problem by itself. Some of the explanations could be found in the original DreamBooth paper. In terms of why it would work for multi-view generation, we think this is because our model is essentially sharing the same modules and weights for 2D and multi-view generation. Thus, the knowledge learned in 2D domain could be potentially transferred to multi-view generation.

---

### Official Review · Reviewer_QGG2 · 2023-10-30

**Soundness:** 3 good
**Presentation:** 3 good
**Contribution:** 3 good
**Rating:** 6
**Confidence:** 3

**Summary:**

The paper proposed a way to finetune large 2D image diffusion models. The goal of finetuning is to allow the model to understand 3D objects.   Thus when we use the finetuned model with SDS loss, we can avoid the multi-face Janus issue. The training set is a mix of 3D multiview data and 2D image data. In the end, the authors show two main applications: 1) finetuning the model with dreambooth; 2) text-to-3d with SDS. The paper showed excellent results.

**Strengths:**

The paper proposed a multiview diffusion model. To reuse existing large 2d image datasets, the model is designed to be able to accept both single images and multiview images as input.
1. The advantage of using both 3D and 2D data is also ablated in the appendix. This is an important observation. It also aligns with our conjecture: even with objaverse, the scale of 3D data is still limited. Using 2D datasets has a significant positive effect when training 3D networks.
2. The trained model will be a valuable asset to this community. The authors proved this by showing how to apply SDS and dreambooth with the method. Thus I encourage the authors to release the rendering program and the training code in the future.

**Weaknesses:**

1. Did the authors consider rendering the images with large elevation? For example, what if we have camera poses at the top or bottom of the object? Will this give more information to score distillation?
2. The main comparison in Fig. 6 is not fair. All other methods are optimization-based by reusing 2d image diffusion. However, the proposed method used additional datasets. A fair comparison would be zero123+SDS?
3. Following the above comment, the trained model should be able to combine with SJC, VDS, and maybe other variants of SDS. This should be discussed somewhere in the main paper or the appendix.
4. Some description of the training algorithm is still confusing. For example,
+ what is the so-called "3d self-attention"? I assume this is done by reshaping FxHxWxC to (FxHxW)xC and then applying self-attention.
+ 32 views are rendered during data preprocessing. How do the authors sample 4 views during training? Uniformly?
+ I hope there will be an algorithm to describe the training process.
5. Optimization time comparison. I would assume the time cost to generative a 3d model is similar to dreamfusion. But maybe with additional 3D information, the optimization would be faster? Or the same?
6. Following the above comment, the method is trained with a large 3d dataset, however, the generating process still relies on an optimization process which is slow. This limits the potential usage of the method.

**Questions:**

See the weakness section.

---

> ### Author Response · Authors · 2023-11-12
> **Response to Reviewer QGG2**
>
> We thank the reviewer for the helpful and careful comments. Here are the response to the reviewer's concerns:
>
>
> 1. **Larger elevation angles**\
> In our initial experiments, we have tried to render camera angles with a wider elevation range. But we found that such diverse camera range leads to more learning difficulty and consequently lower image quality. Therefore, we limit our training data to 0-30 elevations for now. With more data, larger networks and more hyper-parameter tuning, we believe it could be possible to further increase the viewpoint range in the future.
>
> 2. **Zero123+SDS**\
> Yes, we do have compared with Zero123XL+SDS, the results are shown in the appendix A.1 and Figure 10 in the original submission. The basic conclusion is that Zero123+SDS tends to blur and flatten the back of objects if the input image has complex poses or details. In contrast, MVDream can synthesize self-consistent 3D content from the text directly.
>
> 3.  **Combining with SJC/VSD**\
> Theoretically our model is combinable with other SDS variants. For the time limit, we do not have a chance to explore this direction since we think the current SDS results are good enough. We will added such a discussion to the revised paper (Sec 3.2)
>
> 4. **Descriptions**\
> - 3D-attention: yes, that's what it is. We will try to explain it more clearly.
> - Yes. We have explained the sampling of frames in implementation details.
> - We will add an algorithm block
>
> 5. **The time comparison are as follows**\
> We have mentioned a time comparison in the revised paper (Sec 4.2). In short, our method has a similar time consumption as dreamfusion-IF (2 hours) and faster than Magic3D-IF (4h) and ProlificDreamer (10hours).
>
> 6. **Necessity of optimization**\
> We agree with the reviewer that the current optimization process is still taking too long for many use cases. However, as discussed in Section 3.2,  we identify two ways of utilizing a multi-view diffusion model for 3D generation: (1) score distillation and (2) use generated multi-view images for few-shot reconstruction. The latter could be a good strategy for accelerating the overall generation process. But since we do not have such a robust 4-view reconstruction model for the moment, we are not able to demonstrate the results of such a pipeline.

---

### Official Review · Reviewer_nQgK · 2023-10-31

**Soundness:** 3 good
**Presentation:** 3 good
**Contribution:** 3 good
**Rating:** 6
**Confidence:** 5

**Summary:**

The paper proposes a multi-view diffusion model that is able to generate consistent multi-view images based on a given text prompt. Trained on both 2D and 3D data, the proposed method can have generalizability of 2D diffusion models and consistency of 3D renderings. Furthermore, the method can be applied to 3D generation via score distillation sampling.

**Strengths:**

The paper is well-written and easy to understand. The paper focuses on the consistency problem in using 2D diffusion models for 3D generation, which is neglected in previous methods. The proposed method is among the first to adopt the multi-view image generation setting instead of generating views separately. The experiential result seems promising.

**Weaknesses:**

1. How to ensure consistency across multi-view images? There is no particular 3D prior learned in the proposed method.

2. what does re-using 2D self-attention mean? Does it mean inheriting the weights from a pre-trained model or just adopting the same architecture?

3. The description of the training set should be included in the main paper rather than the appendix.

4. Regarding multi-view consistency, it is also important to train the model only on the 3D dataset to see the effect of the utilization of two datasets.

5. In Table 1, are the chosen 1000 subjects from the held-out testing set or a subset of the training set? Moreover, in terms of image quality, it would be better if compared with the current popular method such as zero-123.

**Questions:**

Please see above.

---

> ### Author Response · Authors · 2023-11-13
> **Response to Reviewer nQgK**
>
> We thank the reviewer for the time and helpful comments. Here are our responses to the reviewer's concerns:
>
> 1. **Multi-view consistency** \
> It is true that MVDream does not include any architecture design to inject induct bias to ensure multi-view consistency. The key insight here is that 3D-specific architecture design prohibits the model from share the architecture and weights with 2D image generation models, and consequently limited in terms of generalizability. \
> Therefore, in MVDream, we seek a pure data-driven approach, where we simply re-use image diffusion models for multi-view generation. We believe that this is achievable because dense multi-view generation can be regarded as a special case of video generation, which has already been shown successful. Besides, Zero123 does not have any inductive bias to ensure view consistency either but it is able to generate promising novel views.\
> Once our data-drive approach is proved feasible (which we believe is proved by our generated images), it will be more flexible to be combine with different 2D diffusion models to extend 2D vision prior to 3D.
>
> 2. **Re-using 2D attention** \
> This means (1) we inherit the weights of 2D self-attention layers from Image Diffusion UNet, and (2) during reshaping BxFxHxWxC to Bx(FxHxW)xC for self-attention. (for original 2D models, it was Bx(HxW)xC for self-attention).
>
> 3. **Training Set** \
> We will add the training set description into the main paper.
>
> 4.  **3D dataset only** \
> We have compared with such a model trained on 3D data only, which is discussed in Table 1 in Sec 4.1 and Figure 12 in Appendix A.3.
>
> 5. **Quantitative Evaluation** \
> The evaluation was conducted on a subset from the training set because it was a common practice to calculate FID on training set[1][2]. This being said, we have updated the revised paper by recalculating the metrics on a held-out validation set. We then also evaluated Zero123-XL on the same validation set. But it is worth noting that Zero-123 is an image conditioned model, so the comparison will not be fair. The results are as follows:
> | Method | Condition | FID | IS | CLIP |
> | --- | --- | --- | --- | --- |
> | Zero123-XL | Image |  13.91 | 11.77±0.66 | 26.89±3.25 |
> | Ours | Text |  39.04 | 12.77±0.60 | 30.38±3.50 |
> To evaluate the Zero123 model, we feed one of the ground-truth view (image) of each sample to generate 4 novel views. Because of this, Zero123's results will have a smaller FID, which measures semantic difference between the generated and ground-truth distributions. In contrast, our models shows higher Inception Score and CLIP, which are metrics to evaluate image quality and text-image consistency.
>
>
> [1] Karras, Tero, Samuli Laine, Miika Aittala, Janne Hellsten, Jaakko Lehtinen, and Timo Aila. "Analyzing and improving the image quality of stylegan." In CVPR 2020.
> [2] Ho, Jonathan, Ajay Jain, and Pieter Abbeel. "Denoising diffusion probabilistic models." In NeurIPS 2020.

---

### Meta-Review · Area_Chair_NuJj · 2023-12-05

**Metareview:**

**Summary of the paper**

The paper presents an innovative text-to-3D object generation method using a fine-tuned multi-view diffusion model combining 2D and 3D data. It addresses the multi-face Janus issue through score distillation sampling and DreamBooth-style techniques. The model, trained on both 2D and 3D data, achieves consistency in 3D renderings and generalizability in 2D models, significantly improving over previous methods. Despite complex experiments, it shows promise in generating high-quality, consistent multi-view images from text prompts.

**Strength**

1.The paper is well-written and addresses the consistency issue in 2D models for 3D generation, an area previously neglected. It introduces a novel multi-view image generation approach.

2. The diffusion model uniquely integrates 3D and 2D data, demonstrating the advantages of this combination. The method significantly outperforms current techniques, especially in solving the multi-face Janus problem and ensuring view consistency without content drift.

3. The paper offers clear, intuitive explanations of its methods, produces high-quality results that address common issues, and includes detailed evaluations with various metrics and a user study.

**Weakness**

1. The paper has some confusing parts with missing descriptions in both the algorithm and experiments

2. The submission version lacks several necessary points, including optimization time, hyperparameters, and robustness testing, while most of them were updated during the rebuttal phase.

3. The proposed method seems is limited due to the reliance on a slow optimization process.

**Justification For Why Not Higher Score:**

The presentation is unclear enough, which concerns the reviewers to raise higher scores. Several points are missing, such as optimization time, discussion on combining methods, hyperparameters, as well as robustness testing.

**Justification For Why Not Lower Score:**

The paper is one of the first sets of works that use multi-view diffusion for effective 3D generation, and the method demonstrated is simple and easy to follow. It should be a good contribution to the community.

---

### Decision · Program_Chairs · 2024-01-16

Accept (poster)